# Hierarchical Adaptive Eviction for KV Cache Management in Multimodal Language Models

## Abstract

The integration of visual information into Large Language Models (LLMs) has enabled Multimodal LLMs (MLLMs), but the quadratic memory and computational costs of Transformer architectures remain a bottleneck. Existing KV cache eviction strategies fail to address the heterogeneous attention distributions between visual and text tokens, leading to suboptimal efficiency or degraded performance. In this paper, we propose Hierarchical Adaptive Eviction (HAE), a KV cache eviction framework that optimizes text-visual token interaction in MLLMs by implementing Dual-Attention Pruning during pre-filling (leveraging visual token sparsity and attention variance) and a Dynamic Decoding Eviction Strategy (inspired by OS Recycle Bins) during decoding. HAE minimizes KV cache usage across layers, reduces computational overhead via index broadcasting, and theoretically ensures superior information integrity and lower error bounds compared to greedy strategies, enhancing efficiency in both comprehension and generation tasks. Empirically, HAE reduces KV-Cache memory by 41% with minimal accuracy loss (0.3% drop) in image understanding tasks and accelerates story generation inference by 1.5× while maintaining output quality on Phi3.5-Vision-Instruct model.

## 1 Introduction

Following the notable progress in Large Language Models (LLMs) OpenAI et al. (2024); Touvron et al. (2023); DeepSeek-AI et al. (2024), researchers have begun to investigate the integration of visual information within this framework, culminating in the creation of Multimodal Large Language Models (MLLMs) Bai et al. (2023); Liu et al. (2023a); Abdin et al. (2024). These MLLMs combine visual tokens with textual tokens to enable multimodal interactions within the LLM framework. For instance, Phi-3.5-Vision-Instruct Abdin et al. (2024) integrates a trainable vision encoder–projector pathway that maps image features and text tokens into a shared semantic space, enabling cross-modal autoregressive joint modeling on top of a large language model backbone. Despite the significant advancements in Multi-Modal Large Language Models (MLLMs), the influx of visual and text tokens inevitably incurs substantial memory and computational overhead. This issue stems from the Transformer architecture design, where computational complexity escalates quadratically with token length Vaswani et al. (2017). To mitigate recomputation, the challenge associated with the size of the Key-Value (KV) cache, which stores intermediate attention during generation, has become increasingly critical Pope et al. (2022).

In terms of removing KV cache redundancy, some current methods Li et al. (2024b); Yao et al. (2024); Liu et al. (2024a); Zhang et al. (2024; 2023b); Chen et al. (2024b) mainly focus on the eviction of single-modal tokens and are unable to effectively manage the balance between textual and visual information. For example, SpareVLM Zhang et al. (2024) and MustDrop Liu et al. (2024b) focus on compressing visual tokens to shorten input lengths. However, these strategies mainly tackle redundancy within visual tokens without addressing the interdependencies between text and visual tokens during multimodal long text generation Huang et al. (2025b). H2O Zhang et al. (2023b) and NACL Chen et al. (2024b) remove low-frequency tokens using accumulated attention scores to improve text fluency, but only handle single-modal text tokens. Additionally, these eviction methods require greedy eviction decision calculations at each layer, which limits the model's reasoning efficiency in multimodal understanding tasks. Therefore, there is a pressing need for a robust approach to manage both visual and textual KV redundancy to improve reasoning efficiency and the quality of generation.

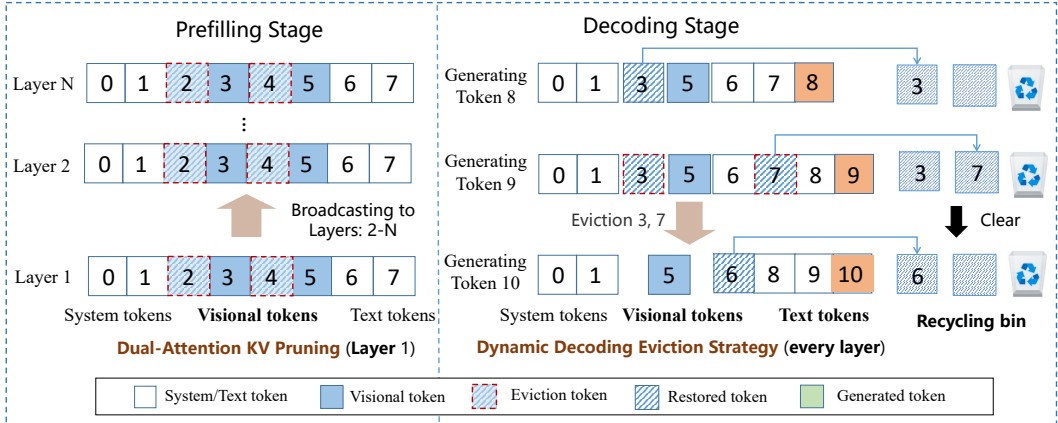

Figure 1: The overall framework of Hierarchical Adaptive Eviction (HAE). In the pre-filling stage, the *Dual-Attention KV Pruning* (DAP) detects redundant visual token indices 2 and 4 in the first layer, and broadcasts this information to perform corresponding eviction across other layers. In the decoding stage, the *Dynamic Decoding Eviction Strategy* (DDES) handles KV eviction. For example, when generating token 9, index 3 is marked but not removed; when generating token 10, index 7 is marked. Once the recycling bin is full, indices 3 and 7 are evicted, and then the indices are reset.

To address this issue, we first conducted experiments to observe the differences in attention distribution between visual tokens and textual tokens during the reasoning process in MLLMs. The observation demonstrates that there is a significant difference in the attention distribution variance between visual tokens and textual tokens. Therefore, redundancy in visual and textual KV cannot be evicted using a uniform eviction strategy. Inspired by this observation, we propose a two-stage eviction framework, named Hierarchical Adaptive Eviction (HAE). The framework (as shown in Figure 1) includes two techniques: *Dual-Attention KV Pruning* (DAP) during the pre-filling stage for the eviction of visual KV redundancy, and *Dynamic Decoding Eviction Strategy* (DDES) during the decoding stage for context-aware eviction optimization in visual-textual mixed contexts.

During the pre-filling stage, we use the DAP method to evict redundant visual tokens in the first layer, and then broadcast the indices of the redundant visual tokens computed from the first layer to the other layers of the model. This strategy is inspired by the experimental observations of the sparsity of attention scores for visual and textual tokens across different layers. The observation results show that the sparsity rate of visual tokens in the first layer is higher than that of textual tokens. Therefore, we specifically evict redundant visual tokens in the first layer to alleviate the pressure on KV storage. Additionally, the sparsity rate of visual information in the other layers of the model is higher than that in the first layer, so we consider broadcasting the indices of redundant tokens to the other layers for eviction. This mechanism has two advantages: a storage advantage, as the KV corresponding to redundant visual tokens is uniformly reduced throughout the network, and a computational advantage, as there is no need for layer-by-layer eviction decision calculations.

During the decoding stage, DDES is used to evict redundant information from the model's KV cache. This strategy is similar to the H2O method, as it determines which KVs need to be evicted by calculating cumulative attention scores. However, unlike the greedy eviction approach (where eviction occurs once per decoding step), DDES uses a recycling bin to hold KV indices associated with tokens that have the lowest cumulative attention scores. Once this bucket is full, the KVs are evicted all at once, similar to a Recycle Bin in operating systems. Compared to greedy eviction that permanently discards KV pairs, DDES enhances efficiency by dynamically retaining potentially relevant KV states. This not only avoids costly recomputations but also ensures the maintenance of model accuracy.

The DAP method can ensure the integrity of modeling information within an allowable error range. We theoretically validated this by establishing an inequality relationship between the number of evicted KVs and the error, as demonstrated in Theorem 2.1. We further obtained through theoretical

analysis that the DDES method has a smaller eviction error during the decoding process compared to the greedy eviction method, as shown in Corollary 2.1.

We conducted a thorough evaluation of the open-source MLLMs (i.e., LLaVA-1.5-7B and Phi-3.5-Vision) through various experiments, including image-based question answering tasks (GQA, ScienceQA, etc.) and long-form story generation (Seed-Story Yang et al. (2024)). The findings highlight the effectiveness of the HAE method. In image-based question answering tasks, HAE reduced KV cache memory usage by 47% while preserving 97% of the baseline model's quality. In image-based story generation tasks, it demonstrated enhanced quality and a 1.5x increase in generation speed compared to the baseline model. In summary, the contributions of this work are three folds:

- A systematic study of cross-modal KV redundancy in MLLMs has been conducted, and a new KV eviction framework, HAE, has been proposed.
- A theoretical framework demonstrates the advantages of preserving KV information integrity, achieving tighter error bounds compared to greedy methods.
- Experiments on multimodal understanding and generation tasks demonstrate our method's superior accuracy, speed, and cache efficiency.

## 2 METHODOLOGY

This section begins with inference experiments to explore the differences in how attention is distributed between visual and textual information, focusing on their sparsity and distribution variance (Refer to Section 2.1). These observations inspired us to design a two-stage KV eviction framework, Hierarchical Adaptive Eviction (HAE) (See Section 2.2). Lastly, a theoretical analysis of HAE is presented in Section 2.3.

### 2.1 OBSERVATION

#### 2.1.1 DIFFERENCES IN CUMULATIVE DISTRIBUTION

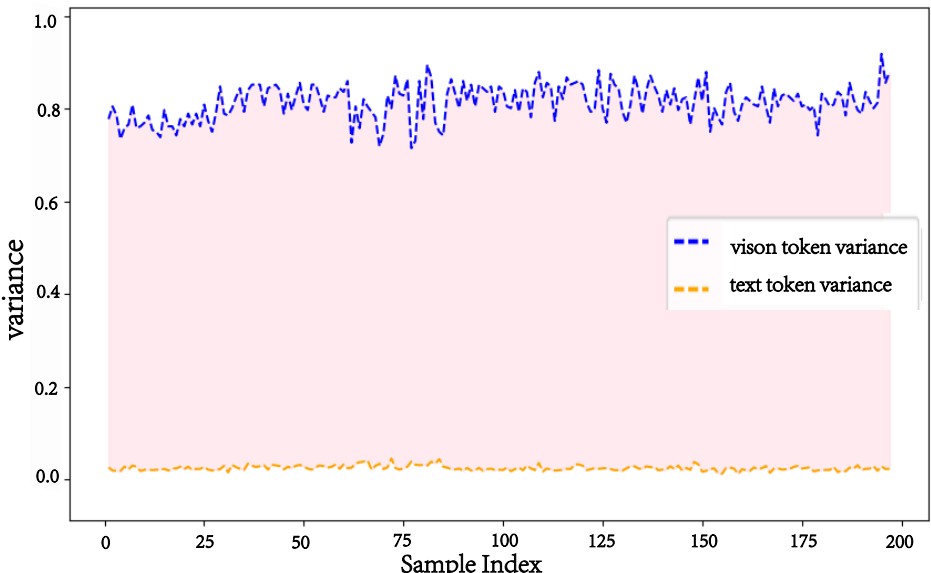

Figure 2: Comparison of cumulative attention score variance between visual tokens and text tokens in the first transformer layer of Phi3.5-vision-Instruction.

The previous works Liu et al. (2024b); Chen et al. (2024a) demonstrated the potential to develop a compact KV-cache size without compromising the efficacy of MLLMs. However, these methods focus solely on evicting one type of KV and overlook the differences in distribution between textual and visual KVs. Creating a redundant KV eviction strategy that considers both modalities will enhance

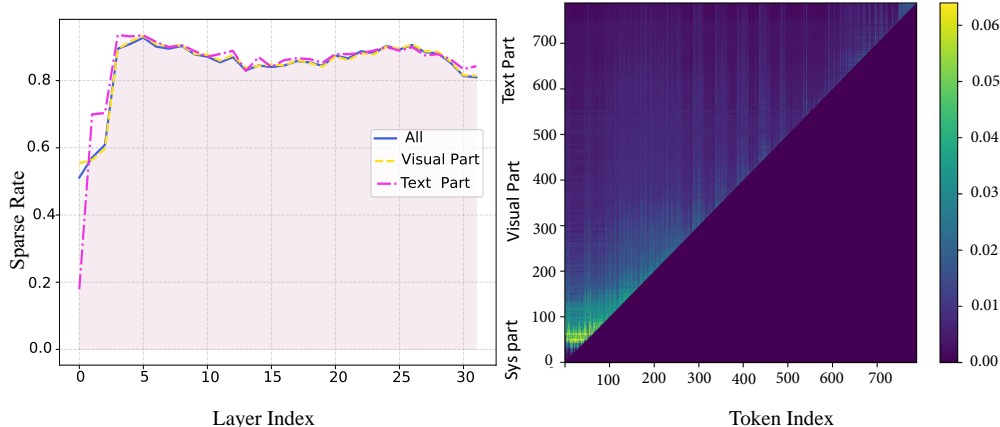

Figure 3: Left: Sparsity rates across layers in Phi3.5-vision-Instruct, highlighting overall, visual, and text component sparsity. Right: Attention matrix for layer 1, featuring annotated token positions that connect visual and text elements to their attention patterns.

the model's efficiency in multimodal understanding and long text generation tasks. To examine the attention distribution differences between the two modalities, we performed a variance analysis on the cumulative attention scores of visual and textual tokens across 200 samples, as illustrated in Figure 2.

**Observation**. As shown in Figure 2, there are significant differences in the distribution of cumulative attention scores for different types of tokens within the attention module. This inspires us to avoid using a uniform eviction decision pattern during KV eviction; instead, we need to employ different KV eviction strategies at different stages.

**Analysis**. Firstly, during the pre-filling stage, it is necessary to determine the types of redundant KV to prioritize for eviction by analyzing the sparsity between output tokens. Additionally, the sparse relationships between visual and textual attention at different layers need to be considered to construct a dynamic redundant KV eviction strategy.

### 2.1.2 SPARSE RATE FOR DIFFERENT LAYERS

Inspired by previous literature Zhang et al. (2023b), we perform zero-shot inference using the pre-trained Phi3.5-vision-Instruction model on the MMMU test set. We plot the attention blocks for each layer and visualize the sparsity rates of the attention matrices for 50 randomly sampled samples, including overall sparsity, sparsity for the visual component, and sparsity for the text component. This result is presented in Figure 3. The code for calculating the sparsity rates can be found in Appendix A.1.

**Observation**. The results in Figure 3 show that although Phi3.5-Vision-Instruction has undergone extensive training, the resulting attention score matrix is still highly sparse. This sparsity indicates that accessing previous key and value embeddings is not always necessary for generating the next token. By observing the visual and textual parts separately, it can be found that the sparsity rate of the textual part is relatively low in the first and second layers, while the sparsity rate of the visual part is relatively high, which is similar to the global sparsity trend.

**Analysis.** Based on a unified eviction strategy, it is not possible to account for the differences in sparsity between text and visual tokens. Therefore, we can consider prioritizing the eviction of redundant visual tokens in the first layer to reduce KV storage pressure. Since the sparsity in other layers is higher than that in the first layer, we can consider reusing the eviction decisions obtained from the first layer in the other layers.

### 2.2 HIERARCHICAL ADAPTIVE EVICTION

In this section, we introduce a two-stage redundant KV eviction framework, as shown in Figure 1. Specifically, during the pre-filling stage, the framework uses Double-Attention KV Pruning (DAP) to

detect redundant visual tokens in the first layer of MLLMs. In the decoding stage, mixed eviction is performed in higher sparsity layers using the Dynamic Decoding Eviction Strategy (DDES).

### 2.2.1 DUAL-ATTENTION PRUNING

To better formalize the description of the DAP method, we first define the eviction process for this stage as follows.

**Definition 1** (*Eviction Policy, pre-filling*). *Let $S_0$ denote the initial ordered set, which consists of the ordered set $V$ of visual tokens and the ordered set $T$ of text tokens. Let $S_1$ denote the target set. During the pre-filling stage, the eviction policy $g:S_0 \to S_1$ meets the following conditions: (1) $|S_0| = n$ (the KV cache size for the inputs); (2) $|S_0 \setminus S_1| < c$ (at most $c$ KV pairs corresponding to the visual tokens are evicted).*

To determine the redundant KV index set, we compute the relevance between visual tokens as well as between each visual token $V_j$ and the global text token. At this stage, the attention scores $A_{i,j}$ between visual token $V_j$ and text token $T_i$ have already been calculated. Inspired by this, the attention mechanism is used to assist DAP in identifying unimportant visual tokens, thereby reducing the storage requirements of the KV cache and improving the model's inference efficiency. We can efficiently obtain the global attention $A_j$ between vision token $V_j$ and all text tokens as follows:

$$A_j = \sum_{i=1}^{|T|} A_{i,j} \tag{1}$$

where $|T|$ is the number of input text tokens.

Then, we can dynamically ascertain the set $V$ of vision tokens, that exhibit the least correlation with the textual content by the threshold $r$:

$$V^p = \{V_j | A_j \geq r \sum_{j=1}^{|V|} A_j\} \tag{2}$$

where $|V|$ is the number of input vision tokens. It is worth noting that the number of visual tokens in set $V^p$ varies dynamically across different test samples rather than being fixed. Therefore, the set $C$ of evicted visual tokens can be defined as $C = V \setminus V^p$.

However, some visual tokens may have a high similarity with individual text tokens, a phenomenon observed in many related works, such as MustDrop Liu et al. (2024b). Therefore, relying solely on this global attention method to filter out insignificant visual tokens might inadvertently exclude tokens essential for sustaining performance but are overshadowed by the overall textual context. To address the issue, we also assess the correlation between each text token and visual token $V_j$. The calculation for individual estimation is as follows:

$$\max_j A_{j,i} < \alpha \tag{3}$$

This means that an evicted visual token must also meet the requirement that its maximum attention score among all text tokens is less than the domain value $\alpha$.

### 2.2.2 DYNAMIC DECODING EVICTION STRATEGY

In this section, we introduce the DDES eviction strategy by formally defining the eviction steps and the conditions that must be met (see Definition 2). This will aid in better understanding and analyzing the advantages of the HAE architecture.

**Definition 2** (*Eviction Policy, decoding*). *Let $S_1 = V^p \cup T$ be the ordered set after eviction in the pre-filling stage, where $V^p \subseteq V$ is the set of retained visual tokens and $|V^p| = |V| - |C|$. The eviction policy during the decoding stage $h : S_1 \to S_2$ satisfies: (1) **Dynamic Cache Constraint**: $l \leq |S_2| < l + D$, where $l = |S_1|$ and $d$ is the buffer threshold for decoding steps. (2) **Recursive Eviction Condition**: For every $k$ new tokens generated ($1 < k \leq D$), an eviction operation is triggered. (3)*

***Cross-Modality importance Criterion***: *Remove the k tokens with the lowest cumulative attention scores from the set, i.e.,*

$$S_2 = S_1 \setminus arg \min_{C \subset S_1, |C|=k} \sum_{C_j \in C} Sc(C_j) \tag{4}$$

*where the scoring function $Sc(c_j)$ is defined as:*

$$Sc(C_j) = \sum_{t=1}^{k} \sigma_j \cdot softmax(\frac{Q_t K_{:j+t}^T}{\sqrt{d}}) + \beta(C_j) \tag{5}$$

*where $\sigma_j$ is a selection function, which selects the $j_{th}$ value from the attention distribution scores. $\beta(C_j)$ represents the cumulative attention score of $C_j$ before the current decoding step. $t$ represents the sequence of tokens generated prior to the current decoding step.*

Our eviction method presents several enhancements. It raises the buffer threshold by $d$ steps compared to the H2O Zhang et al. (2023b) greedy eviction method, offering greater flexibility in token retention. Additionally, it balances the weights of various modal information during the eviction process, unlike existing methods Zhang et al. (2024); Liu et al. (2024b) that primarily target visual token eviction.

## 2.3 THEORETICAL ANALYSIS

To analyze the effectiveness of our proposed HAE framework, we analyzed cache information integrity based on Theorem 2.1 and the error upper bound based on Corollary 2.1. The results of these two aspect analyses help verify the effectiveness of the HAE framework.

**Theorem 2.1** *(Cache Information Integrity). If the eviction threshold $k$ meets*

$$k \leq \frac{log(\epsilon/Attn_{max})}{log(1-\lambda)},$$

*then the total loss of the evicted tokens is $\sum_{j=1}^{c} \epsilon_j < \epsilon$, where $Attn_{max} = min\{\max_j A_{j,i} | j \in [1, 2, ..., c], i \in [1, 2, ..., |T|]\}$, $\lambda$ is the decay rate, and $\epsilon$ is the allowable loss error.*

**Corollary 2.1** *(Error Upper Bound) Let the loss of attention score for each exclusion be $\epsilon_i$. Then the total loss satisfies*

$$\sum_{i=1}^{d} \epsilon_i \leq \sum_{j \in Low_d(S_1)} Sc(C_j), \tag{6}$$

*where $Low_d(S_1)$ denotes the $d$ elements with the lowest scores.*

Theorem 2.1 establishes the relationship between hyperparameters and error during the eviction process. It demonstrates that the HAE method can maintain the integrity of information in the Decoding stage within the allowable error range during the pre-filling stage. Guided by the theoretical insights from the error analysis of H2O Zhang et al. (2023b) greedy eviction in the decoding eviction stage, we derive Corollary 2.1, which states that the error upper bound of dynamic decoding eviction can be smaller than that of greedy eviction. These theoretical results effectively support the effectiveness of the HAE framework. The proof of Theorem 2.1 and Corollary 2.1 can be found in the Appendix A.2.

## 3 RELATED WORK

**Mulitmodal Large Language Models.** Based on the success of Large Language Models (LLMs) such as LLaMA3 Aaron Grattafiori & Jauhri (2024), Phi3.5 Abdin et al. (2024), and Qwen Jinze Bai & Chu (2023) by only receiving text tokens, recent progress has been made in the area of Multimodal Large Language Models (MLLMs) by receiving vision and text tokens. For example, LLaVA Liu et al. (2023b), and Phi3.5-Vision-Instruct Microsoft (2023).

**Efficient Inference of LLMs**. The KV-Cache technology in LLMs is often used to enhance model inference speed Zhou et al. (2024). However, in environments with limited computational resources,

a large amount of KV storage may cause model inference crashes and affect inference speed due to bandwidth limitations Shah et al. (2024); Huang et al. (2025a). Therefore, Dynamic-LLava Huang et al. (2025a), a trainable Sparsification method, have been proposed. These methods necessitate targeted training for adaptation to different tasks. Some methods Zhang et al. (2023b); Chen et al. (2024b) for managing KV-Cache eviction have been proposed to achieve a balance between model inference speed and performance. H2O Zhang et al. (2023b) maintained a balance of the KV pairs corresponding to the most recent and frequently occurring tokens in pure text sequences, which had higher accumulated attention scores. NACL Chen et al. (2024b) optimized the previous strategy of evicting one token at a time during decoding by enabling the eviction of multiple tokens in a single step. However, LLM-based KV optimization methods are not directly applicable to MLLMs because they do not account for distributional differences between modalities.

**Efficient Inference of MLLMs**. To overcome the limitations of KV-Cache storage, previous efforts Zhang et al. (2024); Liu et al. (2024b); Li et al. (2024a) have employed the compression of vision token for MLLMs. Vision tokens are spatial continuous and sparse in semantics compared to dense languages. SparseVLM Zhang et al. (2024) and MustDrop Liu et al. (2024b) dynamically discard low-contribution areas (such as redundant background information) by calculating token attention scores, thereby improving the model's inference efficiency. LLaVA-OneVision Li et al. (2024a) and Dynamic-LLaVA Huang et al. (2025b) can dynamically determine the number of output visual tokens during single image/multiple images/video processing, but these methods require additional training scale.

Our method is a train-free approach that implements different layered expulsions during both the pre-filling stage and the generation stage.

## 4 EXPERIMENTS

We aim to provide experimental evidence for three key research questions: **1**. What advantages in performance and task generalization does HAE have over other eviction methods? **2**. What is the inference time performance of our proposed KV management method, HAE? **3**. What is the rationale behind HAE that enables it to achieve better results? The code and test data are available in the Supplementary Materials.

### 4.1 SETUP

**Models and Tasks** Our experiments are based on three MLLMs, including the Phi3.5-Vision-Instruct, LLaVA-1.5-7B, and Video-LLaVA Zhang et al. (2023a). We validate the HAE method using the LLaVA-1.5-7B model across a diverse set of multimodal benchmarks. This assessment covers various tasks related to multimodal understanding, i.e., image-based challenges, to evaluate the effectiveness of our approach. We conduct experiments on seven widely adopted benchmarks including GQA, MMB, MME, VizWiz, VQA2, TextVQA, and ScienceQA(SQA), all of which are derived from the assessment data of the LLaVA Liu et al. (2023b) model. This part of the experiment used a 4090 GPU 24G. The settings of hardware and hyperparameter can be found in the Appendix A.3.2. More baseline method comparisons can be found in the Appendix A.3.3.

Table 1: Evaluation of Eviction Strategies on Multimodal Understanding Tasks. The best results are highlighted in bold.

| Method | GQA | MMB | MME | VizWiz | SQA | VQA2 | TextVQA |
|---|---|---|---|---|---|---|---|
| LLaVA-1.5-7B (Full cache, 576) | 61.9 | 64.2 | 1862 | 50.0 | 69.5 | 78.5 | 58.2 |
| ToMe (Retain, 192) | 54.3 | 60.5 | 1563 | - | 65.2 | 68.0 | 52.1 |
| FastV (Retain, 192) | 52.7 | 61.2 | 1612 | - | 67.3 | 67.1 | 52.5 |
| SparseVLM (Retain, 192) | 57.6 | 62.5 | 1721 | 50.5 | 69.1 | 75.6 | 56.1 |
| MustDrop (Retain, 192) | 58.2 | 62.3 | **1787** | **51.4** | 69.2 | 76.0 | 56.5 |
| HAE-LLaVA (Ours, Retain, 192) | **61.7** | **64.6** | 1587 | 50.3 | **69.4** | **78.1** | **57.9** |

To demonstrate the advantages of long inference time, we also conducted experiments on image-based long story generation tasks, i.e. Seed-Story Yang et al. (2024), using the Phi3.5-Vision-Instruct base

Table 2: Results of Exclusion Strategies in Multi-Image Story Generation Tasks. $s$ represents the time unit in seconds. The best results are highlighted in bold.

| Method | Style | Engaging | Coherence | Speed($s$) |
|---|---|---|---|---|
| Phi3.5-Vision-Instruct (Full Cache) | 6.65 | 6.60 | 1.82 | 7.40 |
| H2O-Phi3.5-Vision-Instruct | 5.09 | 5.55 | 2.36 | 7.04 |
| MustDrop-Phi3.5-Vision-Instruct | 5.49 | 5.65 | 2.40 | 6.77 |
| HAE-Phi3.5-Vision-Instruct (Ours) | **5.67** | **6.08** | **2.59** | **4.96** |

model. We conducted an experimental evaluation using one RTX4090 24GB GPU. Please refer to the Appendix A.3.2 for hardware configuration information such as CPU and memory. The Seed-Story dataset was constructed aimed at advancing multimodal story generation, which is designed to support the creation of long story sequence. We selected the Rabbids theme from this dataset for our experiments, which includes 332 items, with each item containing 30 images, and each image corresponding to approximately 40-60 English words. For specific details about the dataset and score evaluations, please refer to the Appendix A.3.4.

**Baselines** To ensure a fair comparison, we consider ToMe Bolya et al. (2023), SparseVLM Zhang et al. (2024), FastV Chen et al. (2024a), and MustDrop Liu et al. (2024b) as the baseline method for understanding tasks, and all these methods are designed and tested based on the same base model, i.e., LLaVA-1.5-7B. For image-based long text generation tasks, we use the pure text inference acceleration method, H2O Zhang et al. (2023b) and the MustDrop Liu et al. (2024a) method, which focuses more on visual token-driven approaches, as baseline methods. The baseline methods and our method both use a preset KV-Cache size of 1024 for this task, with a maximum generated text length set to 512. In the ablation experiments, to more thoroughly compare KV eviction methods on pure language models, we also added new baselines, AdaKV Feng et al. (2024) and SnapKV Li et al. (2024c). For a detailed introduction to the baseline methods, please consult the Appendix A.3.1.

## 4.2 MAIN RESULTS

**Multimodal Understanding** We conducted a comprehensive evaluation of the HAE method using multimodal question answering and understanding tasks, comparing it with existing mainstream caching management strategies. All experiments were performed under the same hardware conditions, retaining 192 visual tokens for a fair comparison. The results are shown in Table 1. LLaVA-1.5-7B has 576 visual KV pairs in the full cache scenario. The experimental results indicate that the HAE method performs exceptionally well in multimodal question answering and understanding tasks, approaching the performance of the complete model LLaVA-1.5-7B. Table 4 shows the performance comparison on video understanding tasks, i.e., TGIF Jang et al. (2017), MSVD Xu et al. (2017), and MSRVT Xu et al. (2017). The experimental results indicate that our method achieves the comparable results with the state of the art (SOTA) method, i.e., MustDrop. The results from Table 1 and Table 4 demonstrate that the HAE method can still achieve inference performance close to the original model after removing unimportant tokens, and even surpasses the original model in some tasks.

**Image-based Text Generation** The experimental results in Table 2 indicate that in the image-based story generation task, the HAE method demonstrates advantages in both generation quality and inference speed compared to the baseline models: the H2O method, which focuses on text exclusion, and the MustDrop method, which emphasizes visual tokens exclusion. Compared to the Phi3.5-Vision-Instruct with full cache, HAE achieved scores of 5.67 and 6.08 on the style adaptation (Style) and engagement (Engaging) metrics, respectively, approaching the complete model's scores of 6.65 and 6.60, and significantly outperforming H2O (5.09/5.55) and MustDrop (5.49/5.65). HAE achieves the best overall generation quality balance. In terms of efficiency, HAE's inference speed reaches 4.96 seconds per sample, representing a 33% improvement over the Phi3.5-Vision-Instruct with full cache (7.4 seconds), and significantly outperforms both H2O (7.04 seconds) and MustDrop (6.77 seconds) Appendix A.3.5 provides examples generated by different methods. This validates that HAE's hierarchical exclusion strategy effectively reduces KV-Cache redundancy while retaining cross-modal associative features.

Table 3: Ablation studies on specific strategies are conducted during both the pre-fill and decoding stages. Detailed metrics include the average number of tokens per sample, storage (KV cache), performance (accuracy), and time (average inference time per sample). '*' indicates the average number of tokens retained across all samples.

| Method | Tokens | MMMU Acc. | KV Cache (MB) | Time Sec. |
|---|---|---|---|---|
| Phi3.5-Vision-Instruct | 2357* | 43.0 | 9428 | 0.58 |
| MustDrop (Phi3.5) | 916* | 39.9 | 3664 | 0.56 |
| H2O (Phi3.5) | 956 | 42.8 | 3824 | 0.63 |
| SnapKV (Phi3.5) | 956 | 42.1 | 3824 | 0.59 |
| AdaKV (Phi3.5) | 956 | 42.6 | 3824 | 0.57 |
| HAE-Phi3.5 (Pre-filling) | 1124* | 42.4 | 4496 | 0.21 |
| HAE-Phi3.5 (Decoding) | 956(56) | 42.9 | 3824(224) | 0.49 |
| HAE (All Stage) | 956(56) | 42.3 | 3824(224) | 0.36 |

Table 4: Performance comparison across different methods on various benchmarks. The results include accuracy and score metrics on TGIF, MSVD, and MSRVT datasets. LLM size is 7B.

| Methods | TGIF | | MSVD | | MSRVT | | Avg. | |
|---|---|---|---|---|---|---|---|---|
| | Accuracy | Score | Accuracy | Score | Accuracy | Score | Accuracy | Score |
| VideoChat | 34.4 | 2.3 | 56.3 | 2.8 | 45.0 | 2.5 | 45.1 | 2.5 |
| LLaMA-Adapter | - | - | 54.9 | 3.1 | 43.8 | 2.7 | - | - |
| Video-LLaMA | - | - | 51.6 | 2.5 | 29.6 | 1.8 | - | - |
| Video-ChatGPT | 51.4 | 3.0 | 64.9 | 3.3 | 49.3 | 2.8 | 55.2 | 3.0 |
| Video-LLaVA | 47.0 | 3.4 | 70.2 | 3.9 | 57.3 | 3.5 | 58.2 | 3.6 |
| SparseVLM | 45.7 | 3.2 | 68.2 | 3.9 | 56.0 | 3.5 | 56.6 | 3.6 |
| FastV | 45.2 | 3.1 | 71.0 | 3.9 | 55.0 | 3.5 | 57.1 | 3.5 |
| MustDrop | 46.2 | 3.3 | **71.5** | 3.9 | 56.5 | 3.6 | **58.1** | 3.6 |
| HAE (ours) | **46.8** | 3.3 | 69.7 | 3.9 | **57.0** | 3.6 | 57.8 | 3.6 |

Table 1 and Table 2 demonstrate that our proposed HAE eviction method has certain advantages over other eviction methods in terms of performance and task generalization. Furthermore, in the image-based story generation task (see Table 2), the HAE method shows a significant advantage in inference time for generating longer texts based on image understanding compared to other methods.

## 4.3 ABLATION STUDY

We conducted an ablation experiment on MMMU Yue et al. (2024) to examine the benefits and impacts of specific eviction strategies during the pre-filling and decoding stages. Table 3 reveals the following insights: (1) The H2O method exhibits performance nearly on par with the original model for this task. However, its inference time is longer. This indicates that H2O is not ideal for accelerating multimodal understanding tasks (MMMU), which typically involve generating shorter texts. The time cost from cumulative attention calculations in H2O is not justified in the context of brief text generation typical of MMMU. (2) Although the MustDrop method significantly speeds up inference for MMMU tasks, it focuses solely on removing tokens without considering the differing distributions of cumulative attention scores between visual and text tokens, resulting in a noticeable drop in inference performance. (3) Separate evaluations of token eviction during the pre-filling and decoding stages showed that both achieved inference accuracy similar to the baseline model, with a more substantial improvement in inference speed during the pre-filling stage. The HAE method, which implements eviction in both stages, maintains performance while enhancing inference speed.

In Table 3, restricting eviction to the prefilling phase (HAE-Phi3.5 (Pre-filling)) reduces overall inference time to 0.21s, confirming that DAP significantly cuts initial KV-cache load and speeds up prefilling. Using only the decoding-phase policy (HAE-Phi3.5 (Decoding)) yields 0.49s, showing that DDES's periodic batch eviction (via the recycle bin) outperforms greedy strategies like H2O (0.63s). The full HAE (All Stage) finishes in 0.36s, combining both stages' advantages. It is faster than H2O and MustDrop, demonstrating the synergy of our hierarchical design. Despite improving upon H2O, methods like SnapKV and AdaKV, which excel in long-text generation, still demonstrate lower inference efficiency than our HAE on multimodal understanding tasks.

H2O's inference time can exceed that of the full model in some cases. This stems from its core design: at every decoding step, H2O must compute and sort cumulative attention scores for all cached tokens to decide evictions. Although methods AdaKV and SnapKV have improved upon the H2O approach, they still suffer from the aforementioned issues in multimodal understanding tasks. With a large initial KV cache (due to visual and textual tokens), this frequent sorting accumulates substantial overhead. In short-generation tasks (e.g., MMMU), the cost can outweigh the benefits of cache reduction. In contrast, HAE's hierarchical design reduces computation frequency in two ways: (a) DAP identifies redundant visual tokens only in the first layer and broadcasts eviction indices to all subsequent layers, avoiding per-layer recomputation. (b) DDES introduces a recycle bin that converts high-frequency single-token evictions into low-frequency batch evictions, amortizing cost over multiple steps. Since new tokens still attend to those in the recycle bin, HAE achieves faster inference than H2O while maintaining superior performance.

## 4.4 PRUNING VISUAL TOKEN ANALYSIS

During the model's pre-filling stage, the HAE method reuses the eviction results of visual tokens from the first layer for subsequent layers. While this strategy may seem too aggressive and potentially impact the inference of later layers, we conducted an ablation experiment to analyze whether the positions of the tokens removed in the first layer correspond to those that need to be evicted in other layers. The results are presented in Figure 5, which is shown in Appendix.

We conducted three sets of experiments on the hyperparameter $r$, set to $0.001, 0.0012, 0.0015$ and $0.002$, respectively. This test also helped us select the best threshold hyperparameter, which is $0.0015$. The $\alpha$ is uniformly set to $0.0005$. The results in Figure 4 show the proportion of tokens evicted in the first layer of the Phi3.5-Vision-Instruction model that are also identified as evicted tokens in other layers. Notably, when the global sparse eviction threshold is set to $0.0015$, the proportion reaches its highest point, with an average rate of $90.43\%$. The proportions for the other two threshold values are also greater than $80\%$ in the other layers. This result indicates that the token eviction positions in the first layer can be broadcast to other layers, enhancing the overall inference speed of the model.

## 5 CONCLUSION AND FUTURE WORK

In this paper, we proposed a novel Hierarchical Adaptive Eviction (HAE) framework for efficient KV cache management in MLLMs. The framework reduced memory usage through the development of the Dual-Attention KV Pruning (DAP) technique, maintaining performance levels, and introduced the Dynamic Decoding Eviction Strategy (DDES) to enhance the inference efficiency of MLLMs in generating long texts. Theoretically, HAE guarantees cache information integrity with bounded error propagation. Empirically, it achieves $41\%$ KV-cache reduction while maintaining near-original model quality ($0.3\%$ average accuracy drop), outperforming other methods by $1.5\times$ in inference speed. These results establish a new paradigm for efficient multimodal generation without compromising contextual fidelity.

We identify three promising research paths: *Trainable Eviction*: Developing end-to-end learnable sparsification mechanisms that optimize both attention and KV cache management. *Layer-wise Sparsity Analysis*: Formally analyzing the relationship between sparse attention patterns across transformer layers. This could provide a solid mathematical foundation for our approach and potentially lead to further optimizations. *Large-scale Evaluation*: Conduct comprehensive experiments on extremely large models and datasets to assess the scalability and efficiency of our method in more demanding scenarios.

## STATEMENT

**Ethics statement.** This paper proposes a new Key-Value (KV) cache management framework in vision language models. It cuts KV cache memory with minimal accuracy loss and speeds up inference. We do not identify any potential negative concerns.

**Reproducibility statement.** This paper provides all necessary technique details for reproducibility, including theoretical analysis, algorithm details, experimental settings, and source code of the proposed techniques.

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

# A  APPENDIX

This appendix presents a comprehensive overview of the theoretical proofs, along with reproducible code implementations that include experimental configurations and visualization scripts. Additionally, it details the experimental parameter settings, such as hyperparameters and baseline model versions, and provides supplementary experiments. Together, these elements ensure the research's reproducibility and rigor.

## A.1  OBSERVATION

Figure 3 of the paper shows the sparsity rate of the attention matrices at each layer. The sparsity rate of an attention matrix indicates the proportion of elements in the matrix that are 'close to zero' or 'negligible'. A higher sparsity rate indicates that a larger number of elements within the matrix can be considered negligible, enabling these elements to be omitted during calculations in order to enhance efficiency. The calculation of the sparsity rate here uses a threshold-based sparse computation method, with the following formula:

$$Sparsity\ Rate = \frac{Number\ of\ elements\ A_{i,j} \leq \varepsilon}{Total\ number of elements} \tag{7}$$

where $A_{i,j}$ is an element of the attention matrix $A$, and $\varepsilon$ is the predefined threshold, which is set to $10^{-4}$ when creating Figure 3. The example code for the calculation is as follows:

```python
def compute_sparsity_rate(attention_matrix, epsilon=1e-5):
    # attention_matrix: [batch_size, num_heads, seq_len, seq_len]
    total_elements = attention_matrix.numel()
    near_zero = (attention_matrix <= epsilon).sum().item()
    return near_zero / total_elements
```

## A.2  THEORETICAL RESULT

This section mainly provides proofs and explanations for some theoretical results mentioned in the paper.

**Theorem 3.1** (Cache Information Integrity). If the eviction threshold $k$ meets

$$k \leq \frac{log(\epsilon/Attn_{max})}{log(1 - \lambda)},$$

then the total loss of the evicted tokens is $\sum_{j=1}^{c} \epsilon_j < \epsilon$, where $Attn_{max} = min\{\max_j A_{j,i} | j \in [1, 2, ..., c], i \in [1, 2, ..., |T|]\}$, $\lambda$ is the decay rate, and $\epsilon$ is the allowable loss error.

**Proof**: The proof of the theorem demonstrates that the eviction strategy during the pre-filling stage ensures the integrity of cached information through the definition of the decay model and worst-case analysis.

*Decay Model*: Assume that the attention score of each token $C_j$ diminishes exponentially with time, with a decay rate of $\lambda$. At time step $t$, its score is given by:

$$S(C_j, t) = S_1(C_j) \cdot (1 - \lambda)^t,$$

where $S_1(C_j)$ is the initial score and $t$ is the number of time steps since being cached.

*Eviction Strategy and Loss Calculation*. (1) The first step of the eviction strategy adopts the Dual-Attention Pruning method corresponding to the **Definition 1** provided in this paper and removes $c$ visional tokens.
(2) The total loss is the sum of the scores of all evicted tokens. Let the maximum initial score be $Attn_{max}$, then the loss of a single token at time $t$ when evicted is:

$$\epsilon_j(t) = Attn_{max} \cdot (1 - \lambda)^t.$$

*Worst-case Analysis*. Consider the worst case: every eviction operation swiftly eliminates high-scoring tokens, causing them to remain in the cache for an extended period until they are removed after $k$ evictions. At this stage, the loss for that token is given by:

$$\epsilon_{max} = Atten_{max} \cdot (1 - \lambda)^k.$$

*Total Loss Constraint.* To ensure that the local loss of all evicted tokens does not surpass $\epsilon$, the following must be satisfied:

$$\epsilon \leq \epsilon_{max} => \epsilon \leq Attn_{max} \cdot (1 - \lambda)^k.$$

*Solving for the eviction threshold $k$* taking the natural logarithm of the inequality:

$$\log(\epsilon/Atten_{max}) \leq \log((1 - \lambda)^k),$$

which simplifies to:

$$\log(\epsilon/Attn_{max}) \leq k \cdot log(1 - \lambda).$$

Since $\log(1 - \lambda) < 0$, when dividing both sides by a negative number, we must reverse the inequality:

$$k \leq \frac{\log(\epsilon/Attn_{max})}{\log(1 - \lambda)}$$

**Q.E.D**

**Discussion**. In practical scenarios, the total loss results from the combined impact of various evictions. If the interval between each eviction is $\Delta t$, the total loss can be modeled as the sum of a geometric serious:

$$\sum_{t=1}^{k} Attn_{max} \cdot (1 - \lambda)^k \leq \epsilon.$$

Using the formula for the sum of a geometric series:

$$Attn_{max} \cdot \frac{(1 - \lambda)(1 - (1 - \lambda)^k)}{\lambda} \leq \epsilon,$$

the above inequality holds when $k$ satisfies the conditions of Theorem 3.1.

**Corollary 3.1** (Error Upper Bound) Let the loss of attention score for each exclusion be $\epsilon_i$ . Then the total loss satisfies

$$\sum_{i=1}^{d} \epsilon_i \leq \sum_{j \in Low_d(S_1)} Sc(C_j),$$

where $Low_d(S_1)$ denotes the $d$ elements with the lowest scores.

**Proof**: If the eviction strategy strictly follows the greedy choice (i.e., removing the element with the lowest score currently each time), then the total loss is strictly equal to the sum of the scores of the lowest $d$ elements:

$$Sc(C_j) \sum_{i=1}^{d} \epsilon_i = \sum_{j \in Low_d(S_1)} Sc(C_j) \tag{8}$$

At this point, the upper bound $\leq$ degenerates into an equality. If the $\epsilon_i$ in the Corollary is defined as the loss of HAE strategy, then since there is no urgency to evict tokens during the decoding process as there is with the greedy strategy, the eviction loss for each token corresponding to KV will be smaller. and thus

$$\sum_{i=1}^{d} \epsilon_i \leq \sum_{j \in Low_d(S_1)} Sc(C_j).$$

Further strict proof. By the definition of the greedy strategy, the elements chosen during each eviction operation are always the currently lowest scoring ones in the collection. Therefore, when evicting $d$ elements in a single step, the total loss is directly the sum of the scores of the globally lowest $d$ elements, which is $\sum_{j \in \text{Low}_d(S_1)} Sc(C_j)$.

During stepwise eviction (for example, evicting one element at a time for a total of $d$ times), we prove this through mathematical induction:

- Base Case: In the first step, the lowest scoring element $C_1$ is evicted, resulting in a loss of $\epsilon_1 = Sc(C_1)$.

- Inductive Hypothesis: The loss for the first $k$ steps ($1 < k \le d$) is

$$\sum_{i=1}^{k} \epsilon_i \le \sum_{j \in \text{Low}_k(S_1)} S(C_j).$$

- Inductive Step: In the $k + 1$ step, we evict the new lowest scoring element $C_{k+1}$ from the remaining collection, thus:

$$\sum_{i=1}^{k+1} \epsilon_i \le \sum_{j \in \text{Low}_k(S)} S(C_j) + S(C_{k+1}) = \sum_{j \in \text{Low}_{k+1}(S_1)} S(C_j).$$

Therefore, the total loss is strictly equal to the sum of the scores of the $d$ least scoring elements that have been eliminated.

**Q.E.D**

### A.3 EXPERIMENT

This section mainly provides detailed experimental settings, some experimental evaluation methods, inference examples, and significance tests, among other things.

#### A.3.1 BASELINE MODELS

Below is a basic introduction to the baseline models mentioned in the paper, which can help understand the main objectives of the HAE method and its effectiveness. The baseline models involved include ToMe, FastV, Sparse VLM, MustDrop, and H2O.

- **ToME** Bolya et al. (2023): Token Merging (ToMe) enhances ViT models' throughput without retraining by using a lightweight matching algorithm to merge similar tokens. It works during both inference (doubling throughput with minimal accuracy loss in images, video, and audio) and training (improving training speed, like MAE fine-tuning on video).
- **FastV** Chen et al. (2024a): A plug-and-play method that learns adaptive attention patterns in early layers and prunes visual tokens in later layers.
- **SparseVLM** Zhang et al. (2024): It is a text-guided, training-free mechanism, and uses self-attention matrices of relevant text tokens to evaluate visual token significance, prunes them to maximize sparsity, and employs a rank-based strategy for layer-wise sparsification ratios alongside a token recycling method to compress pruned tokens.
- **MustDrop** Liu et al. (2024b): It evaluates visional token importance across three stages: in vision encoding, it merges similar spatial tokens and retains critical ones; in prefilling, it uses dual-attention filtering guided by text semantics; and in decoding, an output-aware cache policy reduces KV cache size.
- **H2O** Zhang et al. (2023b): The Heavy Hitter Oracle (H2O) reduces KV cache memory by dynamically retaining a balance of recent tokens and Heavy Hitters (H2) — tokens contributing most to attention scores due to frequent co-occurrence.

#### A.3.2 HARDWARE CONFIGURATION AND HYPERPARAMETER SETTINGS

**Hardware Configuration**. This hardware configuration includes an NVIDIA 3090 GPU with 24 GB of VRAM, utilizing driver version 535.183.06, and supports a maximum CUDA version of 12.2. It is powered by an Intel Xeon E5-2682 v4 CPU, featuring 16 cores and 31.4 GB of RAM. The system is equipped with a 20 GB hard disk for storage.

This hardware setup incorporates an NVIDIA 4090 GPU with 24 GB of VRAM, running driver version 535.216.03 and supporting up to CUDA version 12.2. It features an AMD EPYC 7J13 processor with 64 cores and 62.9 GB of memory. The storage configuration includes a 20 GB system disk.

**Hyperparameter Settings**. In this paper, we conduct several experiments to evaluate the performance of our model under different hyperparameter settings. The following table summarizes the hyperparameters used in each experiment.

Table 5: Hyperparameter Settings for Different Experiments. 'RC_size' denotes the size of recycling bin. 'Location' indicates the experimental table corresponding to our methods. '-' indicates that the hyperparameter is not involved.

| Experiment | $r$ | $\alpha$ | n_beams | temperature | RC_size | Location |
|---|---|---|---|---|---|---|
| HAE-LLaVA | 0.0012 | 0.001 | 1 | 0.0 | 64 | Table 1 & Table 5 |
| HAE-Phi3.5 | 0.0005 | 0.0005 | 5 | 0.7 | 128 | Table 2 |
| HAE-Phi3.5 (Pre-filling) | 0.0015 | 0.0015 | 5 | 0.7 | - | Table 3 |
| HAE-Phi3.5 (Decoding) | - | - | 5 | 0.7 | 56 | Table 3 |
| HAE-Phi3.5 (All-Stage) | 0.0015 | 0.0015 | 5 | 0.7 | 56 | Table 3 |

### A.3.3 Image-Based Multimodal Understanding

To provide a more comprehensive comparison, in addition to the fair comparison of the Train-Free method presented in the paper, we also compare it with some of the latest State-Of-The-Art (SOTA) trainable methods. These experimental results are shown in Table 6.

Table 6: Evaluation of Eviction Strategies on Multimodal Understanding Tasks. The "Free" column indicates whether a method is training-free. The best results are highlighted in bold.

| Method | Free | GQA | MMB | MME | SQA | VQA2 | TextVQA |
|---|---|---|---|---|---|---|---|
| LLaVA-1.5-7B (Full Cache, 576) | - | 61.9 | 64.2 | 1862 | 69.5 | 78.5 | 58.2 |
| LLaVA-FastV (Retain, 144) | ✓ | 57.5 | 63.5 | 1459 | 68.7 | 75.1 | 56.2 |
| LLaVA-HiRED (Retain, 115) | ✓ | - | - | - | 74.4 | 74.7 | 44.2 |
| VoCo-LLaMA (Retain, 128) | ✗ | 59.8 | 61.0 | - | - | 76.9 | - |
| Dynamic-LLaVA-7B (Retain, 115) | ✗ | 61.4 | **65.4** | 1480 | **69.1** | **78.0** | 57.0 |
| HAE-LLaVA-7B (Retain, 128) | ✓ | **61.5** | 64.0 | **1497** | 69.0 | 77.5 | **57.1** |

These results demonstrate that the HAE method is a training-free approach that achieves the best performance among training-free methods. Specifically, HAE outperforms other training-free methods such as LLaVA-FastV Chen et al. (2023) and LLaVA-HIRED Arif et al. (2024) across various vision understanding benchmarks. Notably, HAE shows competitive performance, even surpassing some trainable methods (i.e., VoCo-LLaMA Ye et al. (2024) and Dynamic-LLaVA-7B Huang et al. (2025b)) in certain tasks. For instance, HAE scores 61.5 on the GQA benchmark, which is higher than the trainable method Dynamic-LLaVA-7B's score of 61.4. Additionally, HAE scores 77.5 on the VQA2 benchmark, which is slightly lower than Dynamic-LLaVA-7B's score of 78.0 but still very close.

These results indicate that HAE is a promising method with significant potential. Its ability to achieve competitive or even superior performance compared to some trainable methods without the need for training is particularly noteworthy. Although there are benchmarks where HAE does not outperform Dynamic-LLaVA-7B Huang et al. (2025b), such as MMB (64.0 vs. 65.4) and TextVQA (57.1 vs. 57.0), the performance gap is minimal. This suggests that HAE could serve as a strong foundation for developing trainable sparse context sparsification methods in the future. Further research could explore how to build on HAE's strengths to create methods that combine the efficiency of training-free approaches with the adaptability of trainable models.

### A.3.4 Story Generation Dataset

**Dataset Introduction**: SEED-Story StoryStream Yang et al. (2024) is designed for generating captions corresponding to a series of animated images. It encompasses three themes: George, Rabbids, and The_Land. The StoryStream dataset is an innovative resource aimed at advancing multimodal story generation. Originating from popular cartoon series, it includes a comprehensive collection of detailed narratives and high-resolution images, designed to support the creation of long story sequences.

**Sample Description**: The evaluation focuses on the Rabbids theme within the SEED-story StoryStream dataset. This subset comprises 75 image sets, with the validation file (val.jsonl) containing 332 items. Each item consists of 30 images, accompanied by their respective captions. These captions are typically 40-60 English words in length, providing detailed descriptions for each image in the sequence. This structure allows for a comprehensive assessment of story generation capabilities across multiple connected images within the Rabbids cartoon context.

**Prompt for Story Generation**: To use the Phi3-Vision-Instruct model for image-based story generation tasks, we have designed the following prompt to facilitate the reproducibility of experimental results.

Listing 1: Prompt design of the story generation task

```
# Task: Generate a story description for the uploaded image.
# Input: An image.
# Output: A story that meets the requirements.
placeholder = "<|image_1|>\n"
context = (
        "Please generate a story description for the uploaded
            image. The requirements are:\n"
        "- The description for the image should be consistent with
            the style of the 'Rabbids' cartoon, ensuring that the
            text and visuals are aligned in terms of style.\n"
        "- The description should be engaging and entertaining,
            with elements that captivate the audience and maintain
            their interest in the story.\n"
        "- The description should be closely related to the
            content of the image, ensuring that the text is
            coherent with the visuals and maintains logical
            consistency in the narrative.\n"
    )
prompt = f"Image: {placeholder}\nContent: {context}"
```

**Evaluation Metrics**: Three evaluation metrics are used, (1) Style Consistency, (2) Engagement Level, (3) Coherence. The AI judge assigns a score out of 10 to each metric based on precise guidelines for impartiality and structure. The evaluation criteria are outlined in the Listing 2.

Listing 2: Instruct design of different evaluation

```
Style_instruction = "Please act as an impartial judge and evaluate
    the quality of the generation story contents provided by an
    AI assistant. Your job is to give a score out of 10. Your
    evaluation should consider the style consistency of the story
    images. Do not allow the length of the responses to influence
    your evaluation. Be as objective as possible. After providing
    your explanation, output your final score by strictly
    following this format: \"[[ score ]]\", such as \"[[7]]\"."

Engage_instruction =  "Please act as an impartial judge and
    evaluate the quality of the generation story contents provided
    by an AI assistant. Your job is to give a score out of 10.
    Your evaluation should consider the engaging level of the
    story. Do not allow the length of the responses to influence
    your evaluation. Be as objective as possible. After providing
    your explanation, output your final score by strictly
    following this format: \"[[ score ]]\", such as \"[[7]]\"."

Coherence_instruction = "Please act as an impartial judge and
    evaluate the quality of the generation story contents provided
    by an AI assistant. Your job is to give a score out of 10.
    Your evaluation should consider the coherence of the generated
```

␣s t o r y␣i m a g e s␣a n d␣t e x t .␣Do␣n o t␣a l l o w␣t h e␣l e n g t h␣o f␣t h e␣
r e s p o n s e s␣t o␣i n f l u e n c e␣y o u r␣e v a l u a t i o n .␣Be␣a s␣o b j e c t i v e␣a s␣
p o s s i b l e .␣A f t e r␣p r o v i d i n g␣y o u r␣e x p l a n a t i o n ,␣o u t p u t␣y o u r␣f i n a l␣
s c o r e␣b y␣s t r i c t l y␣f o l l o w i n g␣t h i s␣f o r m a t :␣\"[[ s c o r e ]]\" ,␣s u c h␣
a s␣\"[[ 7 ]]\" .

**Evaluation Process**: We used Phi3.5-Vision-Instruct to generate captions for groups of three images at a time. The model is tasked with generating story descriptions that match the style of the 'Rabbids' animated series, maintain high engagement, and ensure coherence with the image content. These generated captions are then compiled into a dataset of 332 items, each containing 30 images and their corresponding captions.

The scoring phase employs the Deepseek-R1 model DeepSeek-AI (2025) to assess the generated content based on three criteria: style consistency, engagement level, and coherence. While style and engagement are evaluated directly using predefined instructions, coherence is assessed by comparing the generated captions to the original ones from the validation file. The scores for each metric are averaged, culminating in a final evaluation score that reflects the overall quality of the generated story content. The score range is from 1 to 10.

### A.3.5 CASE DEMONSTRATION

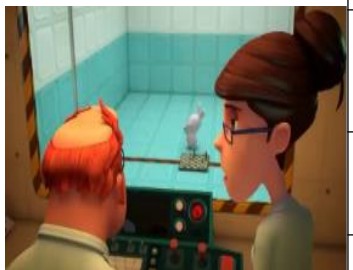

(a)One sunny day in the quirky world of Bunnyland, two scientists were busy at work in their lab, monitoring their latest experiment. They were observing a small, adorable white bunny who seemed to be up to something very interesting.

(b) In the bustling control room of the MegaCorp facility, a pair of diligent workers are engrossed in their duties. One, with a splash of orange hair, is focused on monitoring the monitors, while the other, with their hair tied up neatly, is attentively analyzing the data. Nearby, a robotic figure stands vigilant, its red and black eyes scanning the room. In the background, a futuristic control panel with green and red buttons and lights is in operation, indicating the high-tech nature of this operation.

(c) In the bustling town of Bikini Bottom, where the inhabitants are as colorful as the sky, there's a lively scene unfolding at the local arcade. The arcade, known for its retro video game machines, is alive with the sounds of excitement and the flashing lights of the screens. In the center of this digital playground, a young girl with spiky blonde hair and a bright blue top stands in awe of the glowing arcade machine before her.

(d) In the midst of a bright and airy room, two Rabbid characters find themselves in a curious situation. The first, with a vibrant orange hair and a ponytail that bounces with every move, is engrossed in a peculiar device that seems to be a blend of a keyboard and a game controller. It's not every day you see a Rabbid typing away on such a contraption, but this one is determined to get some work done. The second Rabbid, sporting a neat bun and glasses perched on her nose, watches with a mixture of curiosity and skepticism. Her attention is caught by the screen in front of them, which displays a simple yet intriguing scene.

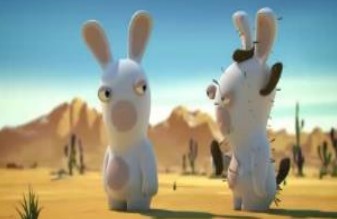

(a) One of the rabbits had landed on a cactus and was now covered in prickly spines. He looked a bit grumpy but unharmed, making the other rabbit chuckle at the sight.

(b) In the vast expanse of the desert, under the scorching sun, two Rabbid bunnies, a regular one and a one-eyed, one-ears, one-tail Rabbid, set out on an epic adventure. The sun casts shadows of the desert's inhabitants behind them, creating a backdrop of eerie yet fascinating silhouettes. The Rabbid bunnies, equipped with their unique features, are on the hunt for the legendary "Golden Carrot of the Desert," a relic said to grant the finder wisdom and power.

(c) In the scorched expanse of the desert, where the sun reigns supreme and the horizon stretches endlessly, two Rabbit friends find themselves on an adventure that's about to turn into a hilarious escapade. They're not just any Rabbids; they are the coolest of the cool, the bravest of the bravest, and the most down-to-earth of the down-to-earth.

(d) In the land of Sunny Grove, two whimsical rabbits, Bunny and Hoppy, were on an epic adventure. The first rabbit, Bunny, had a shiny black rabbit hat that seemed to be a symbol of wisdom or leadership. The second rabbit, Hoppy, sported a black headband that made him look like a cool, laid-back surfer dude. They were out in the vast desert, surrounded by tall cacti and a clear blue sky with the sun casting a warm glow on the scene. They were on a mission to find the legendary Golden Puddle, which was said to grant wishes to those who drank from it.

Figure 4: Comparison of image-based story generation results using different KV eviction acceleration methods based on Phi3.5-Vision-Instruct model. (a) Annotated story description; (b) Results generated using the H2O method; (c) Results generated using the MustDrop method; (d) Results generated using our proposed HAE method.

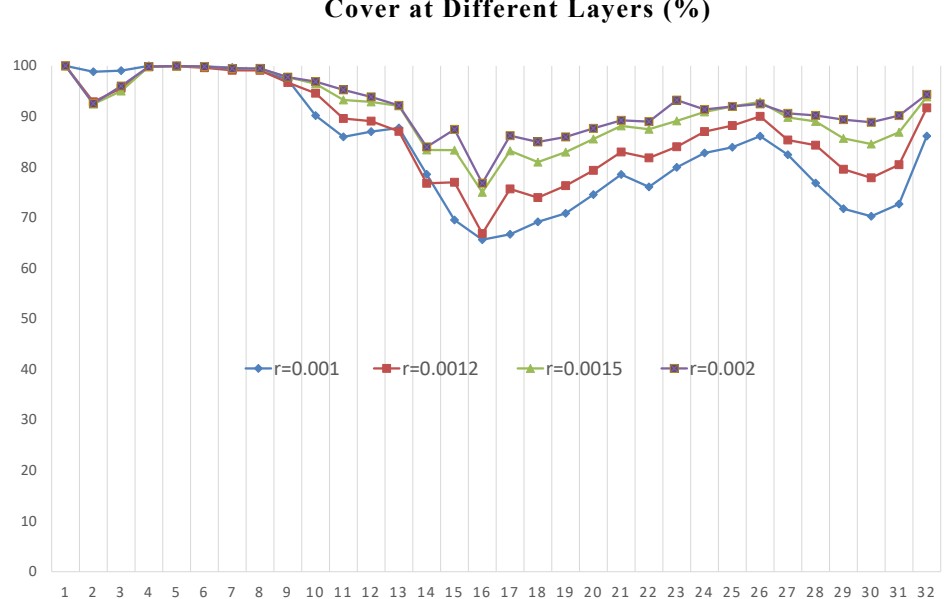

Figure 5: Comparison of cumulative attention score variance between visual tokens and text tokens in the Phi3.5-vision-Instruction.

Figure 4 presents a comparative illustration of image-based story generation results using three distinct KV eviction acceleration methods, i.e., H2O, MustDrop and the proposed HAE, implemented on the Phi3.5-Vision-Instruct model. Each subfigure demonstrates a unique narrative generated from the same input image, highlighting methodological differences.

HAE (d) generates narratives that are precise and tightly aligned with visual content. For instance, it correctly identifies "two Rabbit characters," a "keyboard and game controller," and a "screen," demonstrating strong visual grounding. In contrast, H2O (b) introduces hallucinations like a "robotic figure" and "MegaCorp facility" not present in the image, showing a loss of visual fidelity. Similarly, MustDrop (c) manifests semantic drift with descriptions like "Bikini Bottom" and "scales machine," which are incoherent with the actual scene.

