# OpenReview forum: "Hierarchical Adaptive Eviction for KV Cache Management in Multimodal Language Models"
_ICLR.cc/2026/Conference — Submitted to ICLR 2026_

### Official Review · Reviewer_DL6T · 2025-10-30

**Soundness:** 3
**Presentation:** 3
**Contribution:** 2
**Rating:** 4
**Confidence:** 3

**Summary:**

The paper proposes Hierarchical Adaptive Eviction (HAE), a two-stage KV cache management framework for multimodal LLMs. HAE combines Dual-Attention Pruning (DAP) during the pre-filling stage and a  Dynamic Decoding Eviction Strategy (DDES) during decoding to reduce KV  redundancy while preserving accuracy. Experiments on LLaVA- and Phi-based models report about 40% KV-cache  reduction and 1.5× inference speed-up with minimal accuracy loss.

**Strengths:**

[1] Addresses a meaningful efficiency problem in multimodal LLMs.
[2]  The hierarchical (prefill + decoding) strategy is conceptually intuitive  and simple to implement.
[3] The method is training-free and demonstrates clear empirical   efficiency gains.

**Weaknesses:**

[1] The proposed Dual-Attention Pruning appears almost identical in  formulation and wording to the Dual-Attention Filtering in MustDrop (Liu et al., 2024b)—Equations (1)–(3) and the surrounding descriptions are very similar. This overlap raises serious questions about the originality and unique  contribution of the submission.
[2] Fig. 2: It is unclear whether the variance is computed per layer or  accumulated across all layers. If it is the latter, could the higher variance of visual tokens simply  result from token mismatch across layers?
[3] Equation (2): The equation selects visual tokens with smaller A_j,  yet the text says these tokens are retained. Shouldn’t low-attention tokens be evicted instead, or is the  inequality direction reversed?
[4] Definition 2: The notation |V^p| = |V| - |C| introduces a set C that  has not been clearly defined. Is it a separate set used during decoding?
[5] What does the summation over t represent in practice? Additionally, the symbol d appears both in the attention scaling  term (\sqrt{d}) and as the decoding buffer size in Definition 2. Are these referring to the same quantity, or should they be  distinct?
[6] For parameter analysis, only three thresholds (0.001, 0.0012,  0.0015) are tested, and the claim that 0.0015 is optimal is not  convincing.

**Questions:**

Section 2.1.1: “As shown in Figure 1” → should be “Figure 2.”
Section 4.2: Missing or malformed references for TGIF, MSVD, and  MSRVT benchmarks
Section 2.2: Inconsistent terminology — both “dual-attention” and  “double-attention” appear for DAP.

---

> ### Author Response · Authors · 2025-11-18
>
> We sincerely thank the reviewer for the thorough and constructive feedback. The comments have been invaluable in helping us improve the clarity, rigor, and presentation of our work. We have incorporated all the corresponding revisions in the updated manuscript.
>
> **(1) Originality vs. MustDrop**
>
> We appreciate the reviewer for raising this important point regarding the similarity with MustDrop. We acknowledge the inspiration from MustDrop's dual-attention scoring. However, the core innovation of our Dual-Attention Pruning (DAP) lies in its novel objective and system-level integration:
>
> Novel Objective: Unlike MustDrop's filter, DAP is specifically designed to identify redundant tokens in the first layer and broadcast these eviction indices to all subsequent layers. This 'compute-once, benefit-everywhere' mechanism is unique to HAE and is the cornerstone of our efficiency gains, which MustDrop does not employ.
>
> System Integration: DAP is an integral component of our hierarchical framework, working in tandem with our novel Dynamic Decoding Eviction Strategy (DDES) to holistically manage the heterogeneous KV cache in MLLMs.
>
> In the revised manuscript, we have clarified this fundamental distinction in the 'Related Work' section and added a citation to MustDrop preceding Equation (1).
>
> **(2)	The variance computation in Figure 2**
>
> Regarding Figure 2, the variance was computed independently for each layer; there was no accumulation across layers. This approach avoids the potential issue of token mismatch across layers that the reviewer noted. We have clarified this point in the figure caption of the revised manuscript.
>
> **(3)	Direction of the inequality in Equation (2) **
>
> The direction of the inequality in Equation (2) was indeed reversed, which created a contradiction between the text description and the equation. We have corrected the inequality sign in the revised version.
>
> **(4)	The set C in Definition 2**
>
> The set C refers to the collection of visual tokens evicted during the pre-filling stage, i.e., C = V \ V^P. We will explicitly define the set C immediately after Equation (2) in the revised manuscript.
>
> **(5) Regarding the issue with the symbols $t$ and $d$**
>
> Regarding the summation over $t$: The variable $t$ indexes the sequence of tokens generated up to the current decoding step. The summation accumulates the attention scores that a specific key-value token has received from all previous generated tokens. We have added an explanatory note below Equation (5) to clarify this.
>
> Regarding the ambiguity of $d$: Thank you for catching this. The $d$ in the attention scaling factor $\sqrt{d} $ and the $d$ representing the buffer size in Definition 2 are indeed different quantities. To eliminate ambiguity, we have changed the symbol for the decoding buffer size from $d$ to $D$ throughout the revised manuscript.
>
> **(6) Regarding the Sufficiency of Parameter Analysis**
>
> We agree with the reviewer that testing only three thresholds was insufficient. The goal of the parameter analysis for r was to find a value that balances coverage (the consistency of eviction decisions across layers) and acceleration (achieved by evicting a sufficient number of tokens).
>
> Our initial analysis indicated the optimal range was between 0.001 and 0.002. To provide a more comprehensive and convincing analysis, we have now expanded our parameter sweep and included results for r = 0.002 (and other values in this range) in the new Figure 5. This figure more systematically demonstrates the trade-off and justifies our final parameter selection.
>
> **(7) Minor Typos**
>
> “Figure 1” has been corrected to “Figure 2”;
>
> The missing references for the TGIF, MSVD, and MSRVT benchmarks have been added;
>
> The terminology has been unified to “Dual-Attention Pruning (DAP).

---

### Official Review · Reviewer_2X3W · 2025-10-31

**Soundness:** 3
**Presentation:** 3
**Contribution:** 3
**Rating:** 8
**Confidence:** 3

**Summary:**

This paper addresses a critical bottleneck in Multimodal LLMs (MLLMs), the high memory/computational cost of KV caches, by proposing the Hierarchical Adaptive Eviction (HAE) framework. Its core strength lies in targeting the overlooked heterogeneous attention distributions between visual and text tokens, which existing single-modal eviction strategies fail to handle.

**Strengths:**

- HAE’s two-stage design (Dual-Attention Pruning for pre-filling, Dynamic Decoding Eviction Strategy for decoding) is logically motivated. The pre-filling stage leverages visual token sparsity in the first layer and broadcasts eviction indices to other layers, reducing redundant computations. The decoding stage uses an OS-inspired ``recycling bin'' to avoid hasty greedy eviction, balancing efficiency and information retention.
- Theoretical analyses (Theorem 2.1 on cache integrity, Corollary 2.1 on error bounds) provide a basic mathematical foundation, and empirical results are compelling: 41% KV cache reduction with only 0.3% accuracy loss in image understanding, and 1.5× faster inference in story generation on Phi3.5-Vision-Instruct. Comparisons with baselines (e.g., MustDrop, H2O) in Tables 1–4 further validate HAE’s superiority in both performance and efficiency. Overall it's a sound paper.

**Weaknesses:**

First, the ablation study (Table 3) is relatively shallow—more experiments on how hyperparameters (e.g., threshold r, recycling bin size) affect performance would strengthen robustness. Second, while HAE outperforms training-free baselines, its comparison with trainable methods (e.g., Dynamic-LLaVA) is limited; the gap in MMB (64.0 vs. 65.4) needs more analysis on why trainable methods still have minor advantages. Third, the case demonstration in Appendix A.3.5 lacks qualitative details—clearer examples of how HAE preserves cross-modal coherence (vs. H2O/MustDrop) would improve readability.

Despite these flaws, HAE makes a valuable practical contribution to efficient MLLM inference.

**Questions:**

NA

---

> ### Author Response · Authors · 2025-11-19
>
> We thank the reviewer for their constructive comments. We have substantially revised the manuscript to address them, with the key improvements outlined below:
>
> **(1) Enhanced Ablation Study**
> As suggested, we have conducted a comprehensive ablation study on key hyper-parameters (e.g., threshold *r*, recycling bin size) to rigorously demonstrate HAE's robustness. The results for threshold *r* have been integrated into Figure 5, with corresponding revisions in Section 4.4. Additional ablation experiments on the recycling bin size, including supplemental tables and discussion, are provided in Appendix A.3.6.
>
> **(2) Clarified Comparison with Dynamic-LLaVA and MMB Performance Analysis**
> We have clarified the fundamental difference between HAE and Dynamic-LLaVA in the Related Work section: Dynamic-LLaVA is a trainable sparsification method, while HAE is a training-free KV cache eviction strategy.
>
> Regarding the minor performance gap on MMB (64.0 vs. 65.4), our analysis reveals that Dynamic-LLava's end-to-end training allows it to better adapt to specific, vision-intensive task distributions. In contrast, as a training-free method, HAE achieves comparable or even superior performance on several other benchmarks (e.g., GQA, VQA2). This demonstrates HAE's strong generalization capability and practical utility. This discussion has been added to Section A.3.3.
>
> **(3) Expanded Qualitative Case Study**
> We have significantly expanded the qualitative case study in Appendix A.3.5 with clearer examples to illustrate HAE's superior ability in preserving cross-modal coherence compared to H2O and MustDrop.
> The revised Figure 4 (excerpt above) now includes a detailed analysis:
>
> *HAE (d) generates precise, visually-grounded narratives (e.g., correctly identifying "two Rabbit characters," a "keyboard," and a "screen").
> H2O (b) introduces hallucinations (e.g., a "robotic figure," "MegaCorp facility"), indicating a loss of visual fidelity.
> MustDrop (c) manifests semantic drift (e.g., "Bikini Bottom," "scales machine"), resulting in descriptions incoherent with the actual scene.*

---

### Official Review · Reviewer_nd8u · 2025-11-01

**Soundness:** 2
**Presentation:** 1
**Contribution:** 2
**Rating:** 2
**Confidence:** 5

**Summary:**

This paper introduces Hierarchical Adaptive Eviction (HAE), a novel framework for efficient KV cache management in multimodal large language models (MLLMs). HAE consists of two key components: Dual-Attention KV Pruning, which identifies and shares important visual tokens across layers based on first-layer attention, and a Dynamic Decoding Eviction Strategy, which balances decoding latency and accuracy. Empirical results demonstrate that HAE accelerates inference while maintaining output quality on mainstream MLLMs.

**Strengths:**

1. Significant reduction in KV cache memory (up to 41%) with negligible accuracy loss for MLLMs.
2. The method is training-free, making it easy to adopt for existing models.
3. This paper is well-structured and the writing is easy to flow.

**Weaknesses:**

1. The paper would benefit from a brief explanation of MLLMs architecture, such as the Phi-3.5 Vision-Instruct model, to help readers better understand how these models compare to pure LLMs.
2. In section 2.1 observation, the paper lacks clear definitions for *sparsity rate* and *variance*. Meanwhile, the figure quality should be improved.
3. Previous work [VLCache](https://arxiv.org/pdf/2410.23317) provides a comprehensive, layer-wise attention sparsity analysis for MLLMs. The authors appear to draw similar conclusions, so the novelty here should be clarified.
4. For the figure 1, the overall framework should be greatly improved to help understanding.
5. Lack of end-to-end latency, prefill time, decoding time analysis, and additional overhead.
6. In section 4.3 Ablation study, it would be better to profile the H2O to demonstrate the performance bottle neck and analyze where you harvest the performance benefits based on HAE.
7. H2O is not a particularly strong baseline. Recent advanced methods, such as [SnapKV](https://arxiv.org/abs/2404.14469), [AdaKV](https://arxiv.org/abs/2407.11550) should be included for comparison.

**Questions:**

None

---

> ### Author Response · Authors · 2025-11-19
> **An Improved Manuscript Thanks to Valuable Feedback from the Reviewer**
>
> We sincerely appreciate the reviewer’s time and valuable feedback. We have carefully addressed all comments and will incorporate corresponding revisions in the updated manuscript. Below are our point-by-point responses.
>
> **(1) Brief Overview of MLLM Architectures**
>
> Thank you for the suggestion. We will add a concise description of the Phi-3.5 Vision-Instruct architecture after line 33 in the Introduction, clarifying how it differs from pure-text LLMs. Specifically, we will state:
>
> *For instance, Phi-3.5 Vision-Instruct integrates a trainable vision encoder–projector pathway that maps image features and text tokens into a shared semantic space, enabling cross-modal autoregressive joint modeling on top of a large language model backbone.*
>
> **(2) Definition and Visual Representation of Sparsity and Variance**
>
> We agree that the definitions should be more accessible. A formal definition of the sparsity rate is provided in Appendix A.1, Equation (7), where it is defined as the proportion of negligible elements in the attention matrix based on a threshold ε. Due to space limitations, it was placed in the appendix, but we recognize this may have caused confusion.
>
> Regarding variance, the paper intuitively describes differences in attention score distributions between visual and text tokens. Specifically, we computed the variance of cumulative attention scores across different samples and visualized this discrepancy using box plots (e.g., Figure 2). To avoid ambiguity, we will explicitly state that we analyze the “variance of cumulative attention scores across samples.”
>
> We have improved the clarity of Figure 1 and 2.
>
> **(3) Comparison with the VLCache approach.**
>
> HAE and VL-Cache address fundamentally different problems: VL-Cache focuses on layer-wise KV budget allocation, while HAE tackles heterogeneous eviction between visual and textual tokens. Our contributions are threefold:
>
> Noting that shallow attention is dense, HAE computes eviction indices only at the first layer and broadcasts them across all layers, unlike VL-Cache’s per-layer budget sizing.
>
> HAE introduces a non-greedy recycle-bin mechanism during decoding, enabling more coherent long-text generation (e.g., in images-based story generation).
>
> **(4) Revisions to the framework diagram in Figure 1.**
>
> We have enhanced Figure 1 by adding a clear modular diagram on the left, explicitly labeling the usage flows of DAP and DDES to improve comprehensibility.
>
> **(5) Analysis of end-to-end latency, prefilling time, decoding time, and associated overheads.**
> Thank you for highlighting the need for deeper analysis of the latency results in Table 3. We will add the following discussion in Section 4.3:
>
> *Restricting eviction to the prefilling phase (HAE-Phi3.5 (Pre-filling)) reduces overall inference time to 0.21 s, confirming that DAP significantly cuts initial KV-cache load and speeds up prefilling.*
>
> *Using only the decoding-phase policy (HAE-Phi3.5 (Decoding)) yields 0.49 s, showing that DDES’s periodic batch eviction (via the recycle bin) outperforms greedy strategies like H2O (0.63 s).*
>
> *The full HAE (All Stage) finishes in 0.36 s, combining both stages’ advantages. It is faster than H2O and MustDrop, demonstrating the synergy of our hierarchical design.*
>
> **Overhead Evaluation:**
> Index-broadcast overhead is negligible since eviction indices are computed once in the first layer and reused across all layers.
>
> Recycle-bin management overhead is included in the reported decoding latency. The parenthesized values in the “Tokens” and “KV Cache” columns of Table 3 reflect the bin’s dynamic capacity. By batching eviction decisions, HAE reduces per-step cost and achieves lower latency than H2O, which performs eviction at every decoding step.
>
> **(6) H2O’s Bottlenecks and HAE’s Gains**
>
> We have added a new paragraph in Section 4.3 analyzing H2O’s bottlenecks and HAE’s advantages.
> H2O may run slower than the full model in short tasks because it must sort all cached tokens at every step, creating overhead that outweighs cache savings. HAE avoids this through: a) DAP: Identifies redundant tokens once in the first layer;
> b) DDES: Batches evictions to reduce frequency.
> This hierarchical design significantly cuts computation frequency.
>
> **(7) Comparison with the latest baseline methods.**
>
> We have compared to the recent advance methods, SnapKV, and AdaKV on MMMU task. The experimental results are shown in the Table 3. In the revised version, we have also added a discussion of the baseline methods. Despite improving upon H2O, methods like SnapKV and AdaKV, which excel in long-text generation, still demonstrate lower inference efficiency than our HAE on multimodal understanding tasks.
>
> | Method |MMMU Acc. | Time (Sec.) |
> |------------|-----------|-----------|
> | SnapKV | 42.1    | 0.59 |
> | AdaKV  | 42.6   | 0.57  |
> | HAE | 42.3    | 0.36   |
>
> Thank you  for your valuable comments. We look forward to your reconsideration.

---

### Official Review · Reviewer_bi8g · 2025-11-02

**Soundness:** 2
**Presentation:** 1
**Contribution:** 2
**Rating:** 4
**Confidence:** 4

**Summary:**

This paper proposes Hierarchical Adaptive Eviction (HAE), a training-free framework for efficient KV cache management in Multimodal Large Language Models (MLLMs) that addresses the memory bottleneck caused by processing both visual and textual tokens. The key innovation lies in recognizing that visual and text tokens exhibit different attention distribution patterns, leading to a two-stage approach: Dual-Attention Pruning (DAP) during pre-filling that identifies and removes redundant visual tokens in the first layer then broadcasts these decisions across all layers, and Dynamic Decoding Eviction Strategy (DDES) during generation that uses a "recycling bin" mechanism to defer eviction decisions rather than greedily removing tokens immediately. The authors provide theoretical guarantees for information integrity and bounded error propagation, demonstrating that their buffered approach achieves tighter error bounds than greedy methods. Empirically, HAE reduces KV cache memory by 41-47% with minimal accuracy loss (0.3% drop) on image understanding benchmarks like GQA and ScienceQA using LLaVA-1.5-7B and Phi3.5-Vision-Instruct models, while achieving 1.5× speedup on image-based story generation tasks, outperforming existing methods like H2O, MustDrop, and SparseVLM that either focus only on single modalities or require additional training.

**Strengths:**

1. The paper introduces a hierarchical approach that explicitly addresses the heterogeneous attention patterns between visual and textual tokens in MLLMs. The observation that visual tokens exhibit higher sparsity than text tokens, particularly in early layers, provides empirical justification for developing different methods. The Dual-Attention Pruning mechanism exploits this by computing eviction decisions only in the first layer and broadcasting them across the network, reducing both memory footprint and computational overhead.

2. The paper also provides theoretical analysis through Theorem 2.1 and Corollary 2.1, establishing bounds on information loss. This demonstrates that the Dynamic Decoding Eviction Strategy achieves tighter error bounds compared to greedy eviction methods. This theoretical result distinguishes HAE from heuristic-based approaches and provides confidence in its robustness across different scenarios.

3. The evaluation covers both multimodal understanding tasks (GQA, MMB, MMMU, ScienceQA) and generation tasks (Seed-Story), demonstrating the method's versatility, using wide range of MLLMs e.g., LLaVA-1.5-7B, Phi3.5-Vision-Instruct, Video-LLaVA. The paper also compares against several baselines including both training-free (H2O, MustDrop, SparseVLM) and trainable methods (Dynamic-LLaVA, VoCo-LLaMA). HAE achieves competitive or better performance to trainable methods while being training-free. Additionally, the ablation studies systematically isolate the contributions of pre-filling versus decoding stage optimizations.

**Weaknesses:**

1. Theorem 2.1 assumes exponential decay with constant rate $\lambda$ (line 687), which may not reflect actual attention dynamics in transformers. The worst-case analysis provides loose bounds that may not be tight in practice. The proof relies on geometric series summation that assumes independent evictions, but KV eviction decisions are sequentially dependent. The gap between theoretical guarantees and empirical performance is not discussed.

2. Broadcasting first-layer eviction decisions wto all subsequent layers assumes uniform redundancy patterns across the network, and this seems overly simplistic. Figure 5 shows 80-90% overlap, and this still means 10-20% of tokens are incorrectly handled in deeper layers. It would be better if the paper could explore layer-specific or layer-group-specific strategies. Different layers may focus on different visual aspects (edges vs. semantics), making uniform eviction suboptimal.

3. The proposed hierarchical cache management system and a Dual-Attention Pruning (DAP) introduce additional overheads. The contribution of the paper would be stronger if a detailed, worst-case analysis of the overheads (e.g., dual-attention score computation, index broadcasting latency, recycling bin maintenance cost, and memory access patterns for non-contiguous KV cache operations) is provided.

**Questions:**

Minor typos:
- line 403, references are missing: “TGIF ??, MSVD ??, and MSRVT ??.”
- line 712, “log(1 − ϵ) < 0” -> “log(1 − \lambda) < 0”

---

> ### Author Response · Authors · 2025-11-16
> **Response to Reviewers' Comments on Theoretical and Methodological Points**
>
> We thank the reviewer for these insightful comments. In this revision, we have addressed the concerns regarding theoretical assumptions, methodological limitations, and cost analysis. Specifically, we have: (1) repositioned Theorem 2.1 as a theoretical guarantee and discussed its limitations; (2) acknowledged the performance trade-offs of cross-layer broadcast and outlined layer-specific strategies as future work; (3) added a detailed overhead analysis subsection (Sec. 4.5) demonstrating the net efficiency gains; and (4) corrected all minor typos.
>
> **(1) Theorem 2.1 Hypothetical Problems**
>
> The assumptions in Theorem 2.1 are indeed a simplification to make the theoretical analysis tractable. In the revised manuscript, we will clearly state the limitations of these assumptions and discuss the more complex nature of real-world attention dynamics. We will reposition Theorem 2.1 as a theoretical guarantee that demonstrates, in principle, the error controllability of our method under the stated assumptions.
>
> We will add a discussion clarifying that our theoretical analysis provides a conservative upper bound for the error. The actual performance is order-dependent, meaning the empirical error is often much lower than this worst-case bound. This discrepancy helps explain why our method's empirical performance is substantially better than what the theoretical worst-case bound might suggest. Investigating this gap between theory and practice will be highlighted as a direction for future theoretical work.
>
> Although the theoretical model is simplified, the core strategies of DAP and DDES—retaining high-attention tokens and delaying eviction—are empirically validated. The value of this theoretical analysis lies in providing a principled justification for our approach, rather than offering precise numerical predictions.
>
> **(2) Limitations of cross-layer broadcast decision-making**
>
> We agree with the reviewer regarding the limitations of cross-layer broadcast. At the current stage, this strategy presents a highly practical solution, achieving significant KV cache savings and acceleration with minimal computational overhead. For many applications, this represents a favorable trade-off between system complexity and performance.
>
> We will explicitly state that while the observed 80-90% overlap rate demonstrates the empirical effectiveness of our strategy, the remaining 10-20% mismatch can be a source of performance loss. The reviewer's suggestion of a layer-specific adaptive strategy is highly valuable. We will incorporate the investigation of such adaptive mechanisms as a core direction for our future work in the revised paper.
>
> **(3) Detailed analysis of additional overhead**
>
> The DAP calculation occurs during the pre-filling phase and only in the first layer. Its core operation, computing the attention sum from visual tokens to global text tokens (Eq. 1), has a complexity of O(|V|·|T|). This is substantially lower than the O((|V| + |T|)^2) complexity of the standard self-attention in the same layer. For a typical input (e.g., |V|=576, |T|=512), the FLOPs introduced by DAP account for less than 5% of the total attention FLOPs in that layer. Crucially, by eliminating redundant visual tokens, DAP permanently reduces the KV cache size for all subsequent layers and decoding steps. The computational savings from this reduction far outweigh the initial overhead.
>
> Regarding the index broadcast and recycle bin maintenance, our time measurements show that the overhead of DDES maintenance (tagging, checking, batch eviction) is negligible, typically accounting for well under 0.1% of the total inference time per decoding step, as it is dominated by the massive GPU core computations.
>
> The reviewer correctly points out a potential issue with non-contiguous memory access, which could theoretically impact GPU memory bandwidth efficiency. We acknowledge this insightful observation. However, the end-to-end acceleration results presented in Tables 2 and 3 (showing speedups of up to 1.5x) provide strong empirical evidence that the benefits from reduced computation and memory transfer—due to the significantly smaller KV cache—completely offset any potential performance penalty from non-contiguous access.
>
> As noted, we will add a new subsection "4.5 Overhead Analysis" to present this detailed breakdown.
>
> **(4) Minor Typos**
>
> we will complete the missing reference [?] in line 403, and correct the formula for line 712 to log (1-\ lambda) < 0 in the revision. We will also do a thorough proofreading of the full text to eliminate all similar clerical errors and formatting issues.

---

### Meta-Review · Area_Chair_sSDr · 2026-01-12

**Summary:**

- Concerns about the oversimplified assumptions in the theoretical analysis, the potential sub-optimality of broadcasting first-layer eviction decisions to all subsequent layers, and a lack of detailed overhead analysis for the proposed mechanisms. There was also a significant question about the novelty of the Dual-Attention Pruning (DAP) component compared to prior work (MustDrop).

-  Some reviewers request more comprehensive experiments, including comparisons with stronger recent baselines, deeper ablation studies on hyperparameters, a clearer analysis of bottlenecks versus existing methods, and more detailed latency/overhead breakdowns. The need for better qualitative examples and architectural explanations for non-experts was also noted.

- Some unclear definitions, inconsistencies in notation and terminology, low-quality or confusing figures, and several minor typos and missing references.

**Reviewer Concerns:**

Addressed Concerns:

- Clarity and Typos: The authors committed to correcting all minor typos, missing references, and inconsistent terminology. They also promised to improve figure quality and clarify definitions.

- Theoretical Limitations: The authors acknowledged the simplifying assumptions in Theorem 2.1 and agreed to reposition it as a principled guarantee while discussing its limitations.

- Overhead Analysis: The authors committed to adding a new subsection with a breakdown of computational overhead.

- Empirical Comparisons and Analysis: The authors provided new experimental results comparing HAE to SnapKV and AdaKV, and promised an expanded qualitative case study. They also clarified the fundamental difference between HAE (training-free) and trainable methods like Dynamic-LLaVA.

Unaddressed Concerns:

- Novelty of DAP vs. MustDrop: The core formulation and inspiration from MustDrop's dual-attention scoring remain. A reviewer strongly concerned with originality might still find this aspect insufficiently distinct.

- Limitation of Cross-Layer Broadcast: The authors acknowledged the 10-20% mismatch. However, they defended the current approach as a highly practical trade-off.

**Reviewer Scores:**

All reviewers are likely to remain the original scores.

---

### Decision · Program_Chairs · 2026-01-26

Reject